# Thermal Decomposition of Siderite and Characterization of the Decomposition Products under O₂ and CO₂ Atmospheres

Mariola Kądziołka-Gaweł [1,*], Jacek Nowak [2], Magdalena Szubka [1], Joanna Klimontko [1] and Marcin Wojtyniak [3]

1 Institute of Physics, University of Silesia, 75 Pułku Piechoty 1, 41-500 Chorzów, Poland; magdalena.szubka@us.edu.pl (M.S.); joanna.klimontko@us.edu.pl (J.K.)
2 Institute of Applied Geology, Silesian Technical University, Akademicka 2A, 41-100 Gliwice, Poland; jacek.nowak@polsl.pl
3 Institute of Physics, Silesian University of Technology, Konarskiego 22B, 44-100 Gliwice, Poland; marcin.wojtyniak@polsl.pl
* Correspondence: mariola.kadziolka-gawel@us.edu.pl

**Abstract:** Siderite ($FeCO_3$) is an iron-bearing carbonate mineral that is the most abundant sedimentary iron formation on Earth. Mineralogical alteration of four siderite samples annealed at temperatures 200 °C, 300 °C, 400 °C, 500 °C, 750 °C, and 1000 °C in an O₂ and a CO₂ atmosphere were investigated using such tools as X-ray diffraction (XRD), X-ray photoelectron spectroscopy (XPS), the X-ray fluorescence (XRF) method, differential scanning calorimetry and thermogravimetric analysis (DSC/TGA), and Mössbauer spectroscopy measurements. The decomposition of three siderite samples with similar iron content in the oxygen atmosphere took place in the temperature range of 340–607 °C. This process begins at approximately ~100 °C higher under a reducing atmosphere, but it is completed just above 600 °C, which is a temperature comparable to decomposition in an oxidizing atmosphere. These processes are shifted toward higher temperatures for the fourth sample with the lowest iron but the highest magnesium content. Magnetite, hematite, and maghemite are products of siderite decomposition after annealing in the oxygen atmosphere in the temperature range 300–500 °C, whereas hematite is the main component of the sample detected after annealing at 750 °C and 1000 °C. Magnetite is the main product of siderite decomposition under the CO₂ atmosphere. However, hematite, maghemite, wüstite, and olivine were also present in the samples after annealing above 500 °C in this atmosphere.

**Keywords:** thermal decomposition; iron-bearing carbonate; siderite; Mössbauer spectroscopy; XRD; XPS; XRF; DSC/TGA

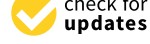



## 1. Introduction

Siderite ($FeCO_3$) is an iron-bearing carbonate mineral that is widespread on Earth and forms in anoxic environments [1–3]. Natural samples of siderite often contain significant substitutions of Mg, Ca, and Mn for Fe in the lattice [4,5], and pure siderite is seldom found. Siderite is the primary mineral, the formation of which governs the ore-forming process [6,7]. Due to its high iron content and light smelting, it plays an important role in steel production [8,9], in petroleum drilling fluids as a scavenger for H₂S, and in processes for making ferrous catalyst materials [10–12] or in the combustion of the coal [13,14]. Siderite is a well-characterized mineral [2,7,15–18], and the thermal decomposition and oxidation of this material have been a topic of considerable interest because of both its commercial importance and scientific questions. It should be noted that it is difficult to upgrade by traditional beneficiation processes such as flotation, gravity concentration, magnetic separation, or any combination of these because siderite is one type of carbonate that is weak magnetically [19]. From an economic aspect, the use of low-intensity magnetic separators is more effective than high-intensity magnetic separators, but they require the modification of



magnetic properties of weakly paramagnetic ores and minerals. The increase in magnetic susceptibility of mineral components can be achieved by phase conversion [20].

The mechanism of thermal decomposition of siderite depends on siderite composition and experimental conditions like temperature, atmosphere, heating rate, or sample microstructure. It has been found that under the influence of oxygen, it is easily oxidized in accordance with the reaction $4FeCO_3 + O_2 \rightarrow 2Fe_2O_3 + 4CO_2$, and hematite ($\alpha$-$Fe_2O_3$) is the final product [21,22]. In a weakly, oxidizing atmosphere, $Fe_3O_4$ was frequently the primary product, and the reactions are expressed using the equation $3FeCO_3 \rightarrow Fe_3O_4 + CO + 2CO_2$ [3]. This process can also go as a two-step process. First, wüstite (FeO) is generated, $FeCO_3 \rightarrow FeO + CO_2$, and then may disproportionate into magnetite ($Fe_3O_4$) in accordance with equation $3FeO + CO_2 \rightarrow Fe_3O_4 + CO$ [23]. Many of the above-mentioned processes of the thermal decomposition of siderite studies have been performed using thermomagnetometry and thermogravimetry [4,5,24,25] or X-ray diffraction analysis [4,26].

The variation in the chemical composition of siderite and different atmospheres used during the annealing process can lead to differences in the results presented in the literature. Therefore, it is necessary to clarify the thermodynamics of siderite decomposition and its kinetics. This also has important implications for processing siderite through a combination of metallurgy and dressing. In this study, the results of the thermal decomposition mechanism of siderite in an oxidizing $O_2$ and reducing $CO_2$ atmosphere and phase analysis of iron-bearing minerals formed during these processes are presented. The Fe-bearing phase transformation and magnetism were characterized using mainly [57]Fe Mössbauer spectroscopy. Mössbauer spectroscopy is a powerful technique for the characterization of the [57]Fe nuclei in iron-bearing materials. We use this method among many other analytical techniques because it can provide information about the [57]Fe hyperfine interactions in various minerals and compounds, which are related to the iron oxidation states, the iron local microenvironments, the iron magnetic states, the relative fractions of iron-bearing components, etc. [27–29]. The Mössbauer investigation was supported by thermogravimetric analyses (TGA) and differential scanning calorimetry (DSC), X-ray diffraction (XRD), X-ray photoelectron spectroscopy (XPS), and the X-ray fluorescence spectroscopy method.

## 2. Materials and Methods

Various carbonate rocks are present in the Upper Silesian Coal Basin, Poland. These are mainly siderites in the form of spherosiderites, shoal siderites, and carbonaceous siderites. The studied siderites come from the upper parts of the Ruda coal seams, in the eastern part of the Chwałowice Trough, in the Upper Silesian Coal Basin. Siderites in the seams occurred in claystone layers, forming intergrowths with a thickness of 0.10–0.80 m. Macroscopically, siderite is gray or dark grey, sometimes with a beige tinge. It has a fine-crystalline structure. The texture is usually massive and chaotic, sometimes directional, and poorly marked as a result of the presence of thin carbon laminae with a thickness of ~0.5 mm (Figure 1). The rocks are cut with carbonate veins 0.5–5 mm thick. These minerals are white in color.

The research was carried out on four rock samples taken from different seams. These samples were designated S1, S2, S3, and S4. All samples were crushed and then ground in a ball mill. After that, powdered samples with a grain size of less than 0.05 mm were obtained. The samples were heated in a muffle furnace at the following temperatures: 200 °C, 300 °C, 400 °C, 500 °C, 750 °C, and 1000 °C. The annealing was carried out in an $O_2$ and a $CO_2$ atmosphere for one hour. There is a time of annealing of samples in which, at high temperatures, there is practically complete decomposition of siderite [22]. Samples annealed in the oxygen atmosphere were placed in trapezoidal crucibles. Cylindrical crucibles with a lid were used for annealing the samples in a $CO_2$ atmosphere. The crucibles were placed in a porcelain vessel, which was purged with $CO_2$ to eliminate the presence of oxygen at the time of filling the crucibles with the sample. After filling the crucibles with samples, they were closed and placed in the furnace.

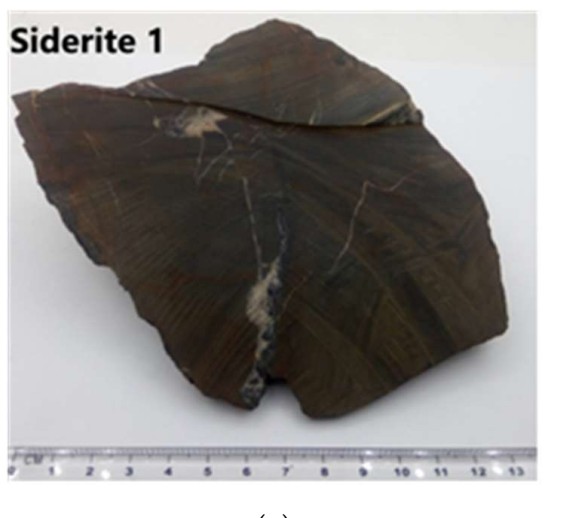 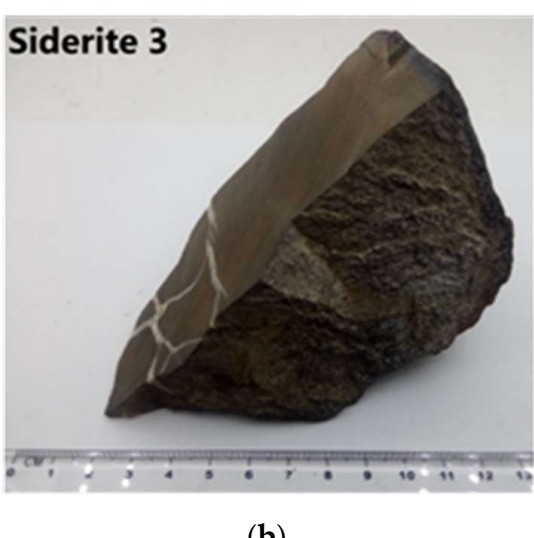

|  |  |
|:---:|:---:|
| (**a**) | (**b**) |

**Figure 1.** Exemplary photographs of the examined siderite samples: (**a**) S1 and (**b**) S3.

### 2.1. X-ray Diffraction

Minerals present in the investigated raw siderite samples and after annealing at 200 °C, 500 °C, 750 °C, and 1000 °C were determined using XRD. These studies were conducted at room temperature by the use of a Siemens D5000 X-ray diffractometer (Siemens, Munich, Germany) and $CuK_\alpha$ radiation. Rietveld refinement was performed in a licensed X'Pert High Score Plus with a PDF-4 crystallography database. The percentages of individual components present in initial samples were determined using the Rietveld method. The content of a given phase should be greater than 2% by volume. It is then assumed that the phase is correctly identified.

### 2.2. X-ray Photoelectron Spectroscopy

XPS spectra were obtained by a PHI5700/660 Physical Electronics Photoelectron Spectrometer with monochromatic $AlK_\alpha$ X-ray radiation (1486.6 eV). The energy of the electrons was measured with a hemispherical analyzer and resolution of about 0.3 eV. Photoelectron emission was measured from a surface area with a diameter of 800 μm and at a take-off angle of 45°. Due to the occurrence of a charge effect, a neutralizer was used for nonconductive samples. To determine the binding energy, the C1s component set at 285 eV was used. Quantification of XPS spectra utilizing peak area and peak height sensitivity factor used the MultiPak Physical Electronics application. Peak shapes were fitted with the Simpeak software. The binding energy values (BE) of each component are presented in the spectrum graph with the estimated error ±0.3 eV. The interpretation of the XPS spectra of the investigated carbonate rocks samples can provide information on the molecular environment, i.e., oxidation state, multiplet structure, chemical bonding, etc., with an error of <10%.

### 2.3. X-ray Fluorescence

The chemical composition of all samples was determined by XRF with a ZSX Primus II Rigaku spectrometer. The spectrometer, equipped with the 4 kW, 60 kV Rh anode, and a wavelength dispersion detection system, allowed for the analysis of the elements from Be to U. No external standards were necessary. Only the internal standards coupled with the fundamental parameters (theoretical relationship between the measured X-ray intensities and the concentrations of elements in the sample) were implemented. The samples for the analysis were prepared in the form of pressed tablets.

*2.4. TGA/DSC Studies*

TGA and DSC were performed using the thermal analyzer NETZSCH Jupiter STA 449 F3. The samples (about 20 mg aliquots of powder) were measured in $Al_2O_3$ crucibles in a definite $O_2$ and $CO_2$ atmosphere. The decomposition process was carried out with a heating rate of 10 K/min under an oxidative atmosphere of oxygen with a flow rate of 25 mL/min between 40 and 1030 °C and under a reducing atmosphere of $CO_2$ with a flow rate of 50 mL/min in the same temperature range.

*2.5. Mössbauer Spectroscopy*

Mössbauer spectra were measured using MS96 spectrometer (Regional Centre of Advanced Technologies and Materials, Palacky University, Olomouc, Czech Republic) operating with triangular velocity reference signal and registration of two 512-channel mirror spectra. Mössbauer spectra were recorded at room temperature in transmission geometry with moving source. A $^{57}$Co(Rh) source with activity of 35 mCi was at room temperature. A multichannel analyzer was used for registration of two mirror spectra in 1024 channels before folding. The spectrometer velocity scale was calibrated at room temperature with a 30 μm thick α-Fe foil. All Mössbauer measurements were carried out on powdered samples. The sample was placed in a plastic container with a diameter of 15 mm and a total window thickness of 2 mm. The plastic window was also designed to absorb the 6 keV X-rays from the beam before they entered the detector. The amount of the sample was chosen to obtain a thickness of ~8 mg Fe/cm$^2$. The evaluation of Mössbauer spectra was performed after folding by least-square fitting using the MossWinn4.0i program [30] with Lorentzian line shape. The spectral parameters such as isomer shift ($\delta$), quadrupole splitting ($\Delta$), quadrupole shift for magnetically split spectra ($\varepsilon$) ($\Delta = 2\varepsilon$), magnetic hyperfine field ($B_{hf}$), line width (full-width at half maximum) ($\Gamma$), and relative subspectrum area (A) were determined. The maximal errors for the Mössbauer parameters were ±0.03 mm/s for $\delta$, ±0.06 mm/s for $\Delta/2\varepsilon$, ±0.2 T for $B_{hf}$, ±0.06 mm/s for $\Gamma$, and ±1 % for A. The values of $\delta$ are given with respect to α-Fe at 295 K.

**3. Results**

*3.1. X-ray Diffraction*

Measurements by XRD revealed that siderite samples contained siderite as the main component and small amounts of accessory minerals (Table 1). Kaolinite was detected in all samples and dolomite in samples S2 and S4. Trace amounts of quartz and illite were present in all samples. For sample S3 the concentration of quartz was the highest (~7%).

**Table 1.** Mineral composition of investigated siderite sample based on XRD analyses by Rietveld method in vol.%.

| Component | Formula | Sample | | | |
|---|---|---|---|---|---|
| | | S1 | S2 | S3 | S4 |
| Siderite | $FeCO_3$ | 94.2 | 90.3 | 83.6 | 85.5 |
| Quartz | $SiO_2$ | Trace | Trace | 7.2 | 0.3 |
| Illite | $K_{0.65}Al_{2.0}[Al_{0.65}Si_{3.35}O_{10}](OH)_2$ | 1.0 | 1.2 | 3 | 0.9 |
| Kaolinite | $Al_2Si_2O_5(OH)_4$ | 4.8 | 2.2 | 6.2 | 0.2 |
| Dolomite | $CaMg(CO_3)_2$ | Trace | 6.3 | Trace | 13.1 |
| | sum | 100 | 100 | 100 | 100 |

X-ray diffraction studies made it possible to trace the phase composition changes that occur during the annealing of the samples at various temperatures and in different atmospheres ($O_2$ and $CO_2$), and phases analysis of iron oxides formed during these processes are presented. Figure 2 shows, as an example, the diffractograms obtained for sample S1 annealed at temperatures 200 °C, 500 °C, 750 °C, and 1000 °C in these external conditions.

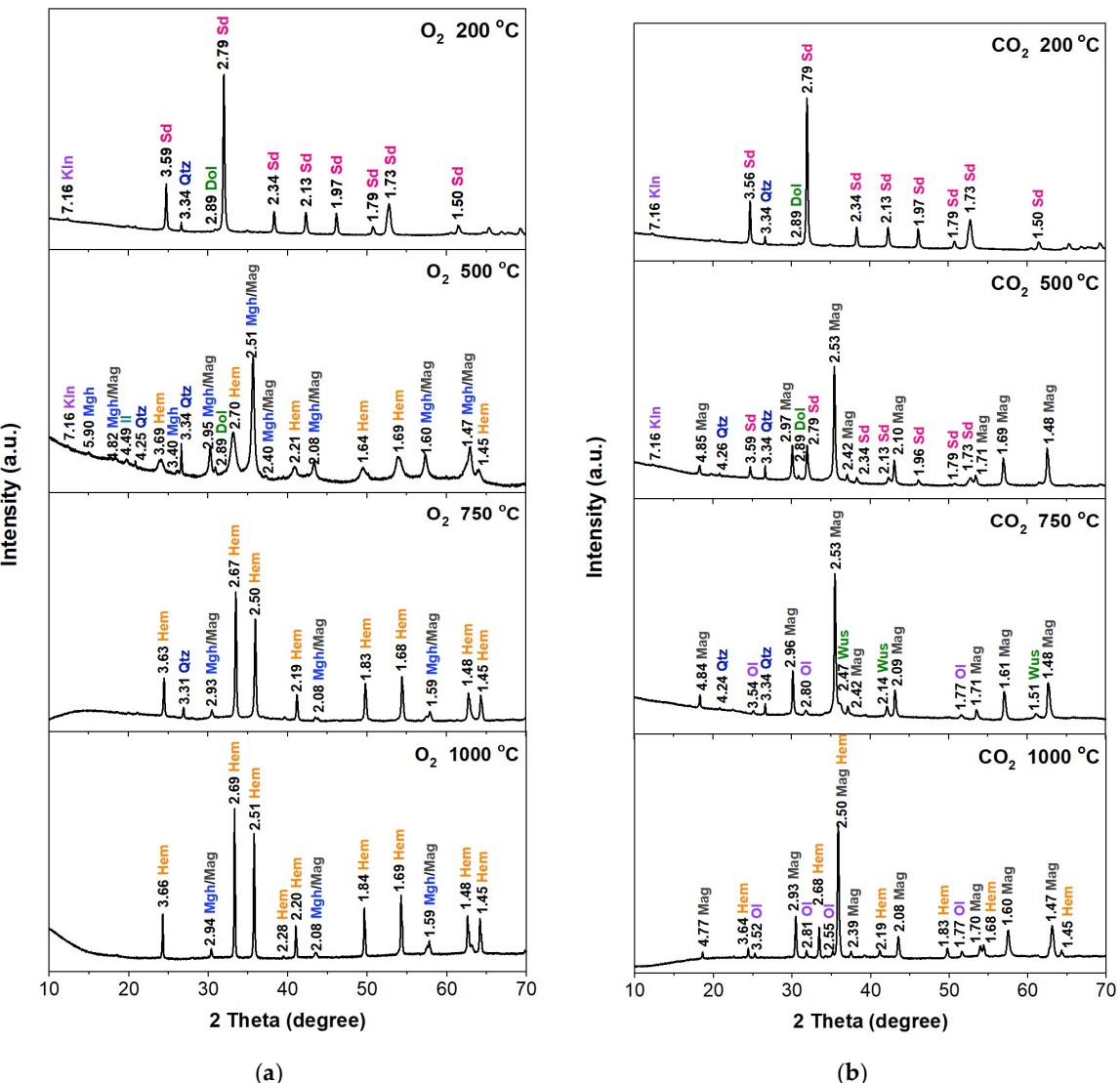

**Figure 2.** XRD patterns of the S1 sample annealed at different temperatures in the O₂ (**a**) and CO₂ (**b**) atmosphere. The interplanar spacing values are shown in the diffractograms. Sd—siderite, Kln—kaolinite, Dol—dolomite, Mag—magnetite, Mgh—maghemite, Hem—hematite, Il—illite, Qtz—quartz, Ol—olivine.

Samples annealed in the oxygen atmosphere at 200 °C did not show changes in the phase composition. Annealing the samples at 500 °C caused changes in the phase scale. Siderite was no longer present in the sample, but hematite ($\alpha$-$Fe_2O_3$), a product of the thermal decomposition of siderite in an oxygen atmosphere, was present. Reflections originating from maghemite ($\gamma$-$Fe_2O_3$) and/or magnetite ($Fe_3O_4$) were also found. On the diffractograms, the reflections from these minerals coincide with each other. After annealing the samples at 750 °C, hematite was found to be the dominant component in the sample. Reflections that may come from maghemite or magnetite were also found, but their intensities were much lower than in the sample annealed at 500 °C. The diffractograms of the samples heated at 1000 °C showed no significant changes in relation to the samples heated at 750 °C. Samples annealed in a CO₂ atmosphere at a temperature of 200 °C, similar to samples annealed in the oxygen atmosphere at this temperature, did not show changes in the phase composition. Samples annealed at 500 °C in a CO₂ atmosphere showed a different phase composition compared to the initial samples. In samples S1, S2, and S4, the dominant phase is magnetite ($Fe_3O_4$), which was formed as a result of thermal transformations of

siderite. The diffractograms also show clearly marked siderite reflections. In sample S3, the dominant phase is still siderite, but reflections from magnesioferrite were also found.

The admixtures of quartz and some clay minerals probably slowed down the transformation of siderite into magnetite in this sample. After annealing the samples at 750 °C in a $CO_2$ atmosphere, siderite reflections were no longer observed on the diffractograms. Only the presence of magnetite and wüstite (FeO) was found. Small amounts of quartz were also present. The presence of some reflections, which may indicate the formation of minerals from the olivine group (($Mg^{2+}$,$Fe^{2+}$)$_2$$SiO_4$), was also found. After annealing the samples at 1000 °C, the diffractograms revealed the absence of wüstite and quartz, except in sample S3. The diffractograms were dominated by magnetite and hematite. Reflections similar to olivine were also found.

### 3.2. X-ray Photoelectron Spectroscopy

XPS spectra obtained in the region of the Fe2p, C1s, O1s, and K2p core levels for all investigated carbonate rock samples are displayed as a plot of electron binding energy vs. intensity for initial samples and are shown in Figure 3. The binding energy values corresponding to the XPS spectra components are presented on the graphs. The Fe2p spectra are characterized by a doublet structure relating to spin-orbit splitting between $2p_{1/2}$ and $2p_{3/2}$ states. In addition to the multiplet structure, the shake-up satellite peaks are present, arising from intrinsic energy losses when the photoelectron leaves the hosting atom. The Fe2p spectra of all initial siderite samples appear similar regarding the position and shape of the peaks (Figure 3a). The binding energy of the $Fe2p_{3/2}$ peak is 710.6 eV, and that for $Fe2p_{1/2}$ is 724.1 eV. The $Fe^{2+}$ satellite lines are localized at 715.2 and 728.7 eV. These energies are assigned to $Fe^{2+}$ ions in $FeCO_3$ [31,32].

The O1s core-level XPS line of the initial samples (Figure 3b) comprises two separate peaks at 530.2 and 532.3 eV binding energies, which can be related to different forms of oxygen bonding. The most intense line at 532.3 eV is ascribed to the O species in carbonate [29,30] or, more generally, to C-O (oxygen singly bonded to aliphatic carbon). The second peak with low intensity is typical for the presence of oxygen–metal bonds [33]. Potassium K2p has spaced spin-orbit components (Figure 3b) whose binding energies are 293.6 and 296.4 eV, which are probably related to $K^+$ ions in illite. The intensities of these doublet lines, compared to the intensity of the C1s line, are the highest for sample S3 than for the S1 sample. For S2 and S4, the line intensities are the lowest, which can reflect the concentration of illite in the samples. The XPS spectra for the carbon element (C1s) contain a peak at 285.0 eV, which is typically assigned to adventitious (aliphatic) carbon found in all samples exposed to the atmosphere, and a peak centered at 289.8 eV corresponding to carbonate in siderite [33,34].

### 3.3. X-ray Fluorescence

Figure 4 presents the contents of the elements in the investigated siderite samples. These plots are in regard to elements whose concentration was higher than 0.1 wt.%. Of course, the analyzed materials contained elements like S, Cl, Ti, Cr, Ni, Zn, Sr, or Ba, but their total concentration did not exceed 0.2% of the total weight. Samples S1, S2, and S4 contain a similar concentration of Fe (~36.5 wt.%), Si (~4.2 wt.%), Al (~2.5 wt.%), Mg (~1.1 wt.%), and Ca (~1.2 wt.%), which, similar to O (~51.7 wt.%) and C (~7.1 wt.%), are elements of minerals present in investigated siderite samples. Sample S3 contains about 10% less Fe (~27.7 wt.%) in comparison to other samples and also contains three times more Mg (~4.3 wt.%). The iron content in all tested samples is about 37%, except in sample S3, where iron constitutes about 28% of the sample's volume (Figure 4) and contains much more Mg than other samples. The concentration of this element increases above 40% for all samples (unchanging for S3) after annealing at 500 °C and above 50% after annealing at 1000 °C. For S3, iron constitutes about 35% of the total volume.

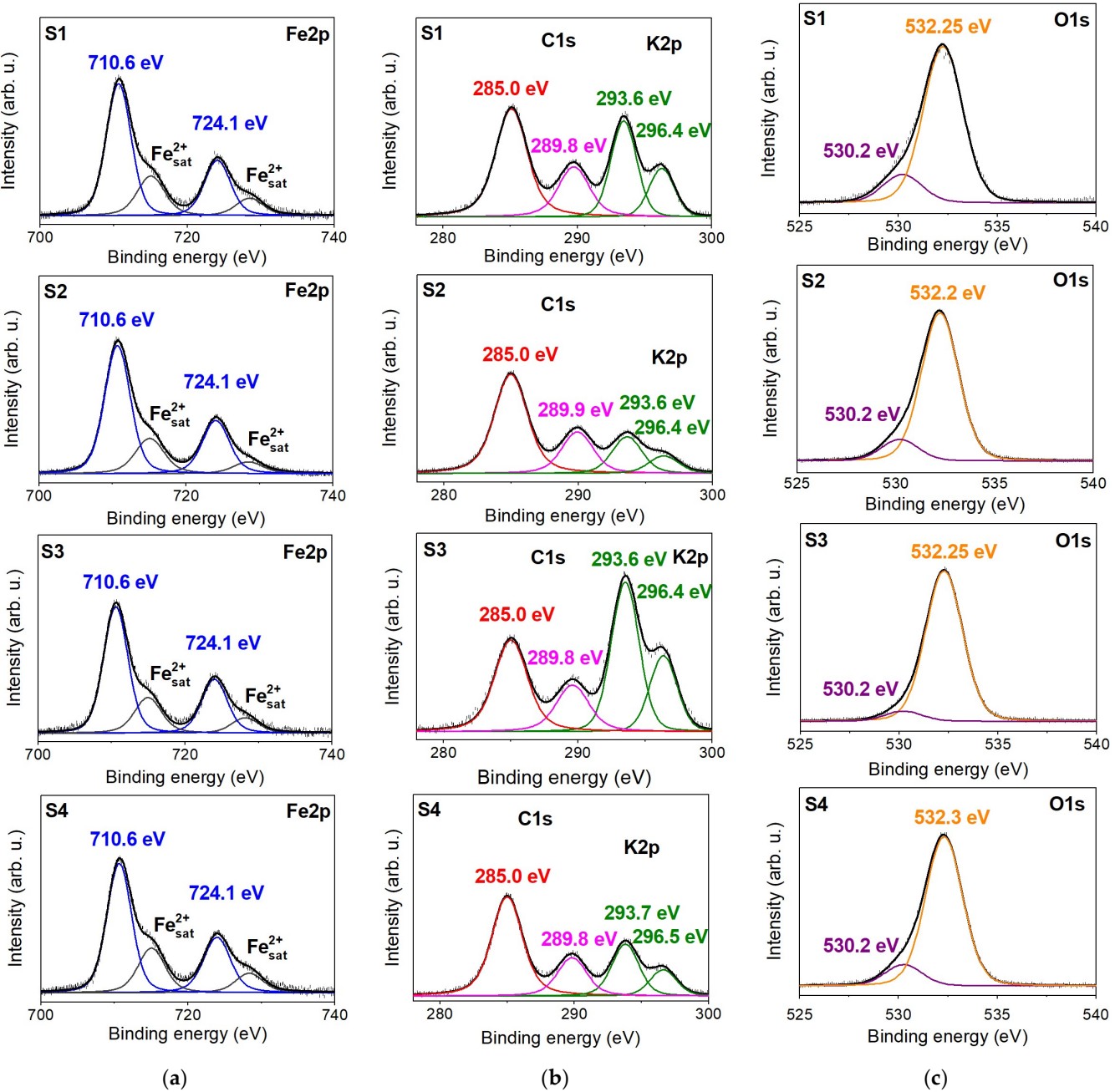

**Figure 3.** The XPS spectra showing the (**a**) Fe2p, (**b**) C1s and K2p, and (**c**) O1s regions with relative peak deconvolutions obtained for the investigated siderite rock samples (S1, S2, S3, and S4). The binding energy value of each component is present on the graph.

### 3.4. TGA/DSC

The TGA and DSC profiles for the thermal degradation of siderite samples obtained in an oxidizing atmosphere and under a reducing atmosphere ($CO_2$) are shown in Figure 5. Table 2 contains detailed values of temperature ranges where processes of dehydration and decomposition of siderite present in investigated samples take place. Throughout the decomposition process of siderite, a small mass loss (<3%) appeared in the temperature ranges: 40–340 °C in the $O_2$ atmosphere and 40–447 °C under the $CO_2$ atmosphere, corresponding to a tiny endothermic peak, which was most likely due to the moisture evaporation in the samples.

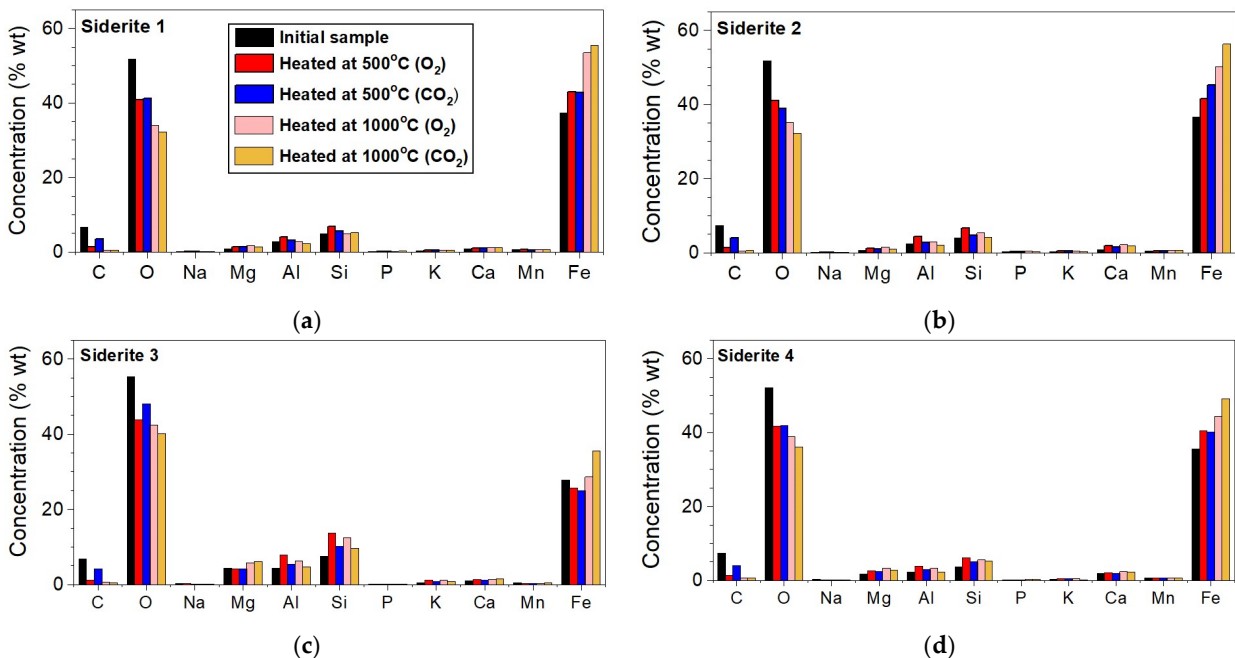

**Figure 4.** Elemental concentrations for samples annealed at 500 °C and 1000 °C under $O_2$ and $CO_2$ atmospheres of the siderite 1 (**a**), siderite 2 (**b**), siderite 3 (**c**), and siderite 4 (**d**).

**Table 2.** Temperature ranges of mass losses for investigated siderite samples estimated from TGA/DSC curves.

| Atmosphere | Sample | Dehydration | | Decomposition | | |
|---|---|---|---|---|---|---|
| | | T Range, °C | ΔT °C | T Range, °C | ΔT, °C | $T_{max}$, °C |
| $O_2$ | S1 | 40–340 | 300 | 340–582 | 242 | 466 |
| | S2 | 40–342 | 342 | 342–576 | 234 | 455 |
| | S3 | 40–330 | 290 | 330–653 | 323 | 570 |
| | S4 | 40–338 | 298 | 338–607 | 269 | 454 |
| $CO_2$ | S1 | 40–434 | 396 | 434–577 | 143 | 501 |
| | S2 | 40–437 | 397 | 437–541 | 110 | 493 |
| | S3 | 40–447 | 447 | 487–617 | 130 | 573 |
| | S4 | 40–425 | 385 | 425–602 | 177 | 474 |

Decomposition of siderite samples in the oxygen atmosphere took place in the temperature range of 330–653 °C (Table 2) with a maximum rate at 466 °C, 455 °C, 570 °C, and 454 °C, respectively, for S1, S2, S3, and S4. The initial decomposition temperatures of the studied siderites were very close to 350 °C observed for pure siderite [35]. A significant endothermic peak was observed in this temperature range (Figure 4) due to the intense endothermic decomposition reaction of siderite. Siderite dissociation follows the formula: $FeCO_3 \rightarrow FeO + CO_2$, which causes an endothermic effect on the DSC curve (in the case of the S1 sample this is with a maximum temperature of 406 °C). Almost immediately, the following oxidation reaction is superimposed on it: $2FeO \rightarrow Fe_2O_3$, which is manifested by exotherms (for S1 the maximum is at 459 °C). This reaction is usually associated with a slight increase in the mass of the sample, but this increase in the tested samples is not detectable on the TG curve. The total mass loss in this temperature range was about 29.5%

for all samples. The thermograms of samples S2 and S4 still show slight endothermic effects and peaks in the DTG curves with a maximum of about 700 °C. They are related to the presence of dolomite in these samples [36].

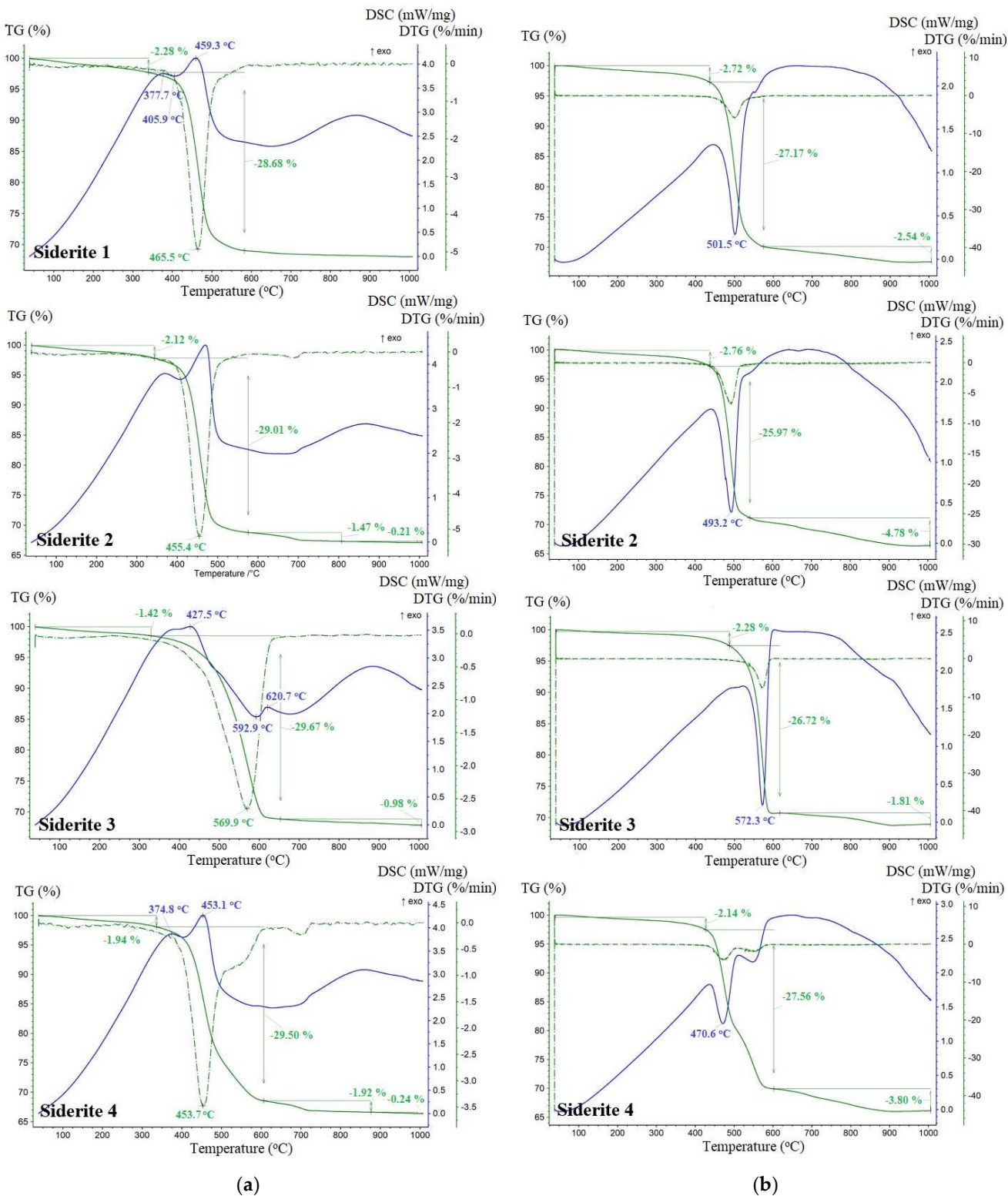

(**a**)　　　　　　　　　　　　　(**b**)

**Figure 5.** TGA, DTG, and DSC curves of investigated siderite samples under $O_2$ (**a**) and $CO_2$ (**b**) environment.

### 3.5. Mössbauer Spectroscopy

Obtained Mössbauer spectra of all non-heated siderite samples were characteristic of a quadrupole doublet. This doublet reflects the presence of $Fe^{2+}$ ions in the siderite structure. The values of $^{57}Fe$ hyperfine parameters of these doublets are the same: $\delta = 1.23–1.24$ mm/s and $\Delta = 1.80–1.81$ mm/s for all samples (Table S1, Supplementary Materials) and agree well with the literature data [37–39]. The absence of other components in the Mössbauer spectra indicates that siderite is the only iron-bearing mineral in the studied samples.

Mössbauer spectra of siderite samples annealed at temperatures from 400 °C to 1000 °C in different atmospheres ($CO_2$ and $O_2$), together with fitted subspectra, their assignment and relative areas are presented in Figures 6–9. The Mössbauer parameters related to all fitted components are listed in Table S1. Additionally, Table 3 contains Mössbauer parameters of the fitted subspectra obtained for sample S1 annealed at different temperatures and atmospheres. The spectra of all samples annealed at 200 °C remain unchanged regardless of annealing conditions and contain only a ferrous doublet related to siderite (Table S1, SM). The observed process of thermal decomposition of siderite looks very similar for samples S1, S2, and S4 but differs from that observed for the S3 sample. The decomposition process of siderite annealing in the $O_2$ atmosphere starts at 300 °C. Although the doublet related to siderite constitutes the main component of Mössbauer spectra, ferric doublet and sextets are also present (Table S1, Table 3). The relative area (roughly the iron fraction) of the ferric doublet is about 15% for samples S1, S2, and S4 and ~4% for sample S3. The $\delta$ values of this doublet are in the range 0.31–0.43 mm/s and the $\Delta$ values are in the range 0.88–1.07 mm/s. Such hyperfine parameters can be related to iron-oxide nanoparticles (NPS) [40–42]. To be sure that this doublet is related to the NPS, the S1 sample annealed at 400 °C was measured at the liquid nitrogen (LN) temperature. As the temperature decreases, the contribution of the paramagnetic component associated with NPS decreases while the contribution of the sextet associated with hematite increases (Figure S1 and Table S2, SM). The relative area of this doublet increases with increasing annealing temperatures up to 500 °C.

Magnetic sextet with $\delta = 0.37$ mm/s, $\varepsilon = -0.15$ mm/s, and $B_{hf} = 51$ T is related to hematite [43,44]. After annealing the samples at 400 °C, this sextet makes up the main component in the Mössbauer spectra instead of siderite ferrous doublet. Beyond that, the appearance of two magnetic sextets with $B_{hf}$ ~47 T and ~44 T, respectively, corresponds to $^{57}Fe$ nuclei at the tetrahedral (A) and octahedral (B) sites in magnetite ($Fe_3O_4$). At room temperature, the A-site sextet is characterized by a relatively low $\delta$ value of ~0.28 mm/s and an indeterminately low $\varepsilon$ value. The $\delta$ value of the B-site sextet is larger and determined by the electron hopping $Fe^{3+} \rightarrow Fe^{2+}$. With respect to the lifetime of the $^{57}Fe$ in the excited state, this leads to the formal $Fe^{2.5+}$ valence in the B sites. The $\delta$ value for the B-site sextet observed at room temperature is about 0.63 mm/s and, like the A site, has a low $\varepsilon$ value [44]. It also should be noted that, usually, the room temperature Mössbauer spectra of magnetite were decomposed using two magnetic sextets assigned to $^{57}Fe$ in the A and B sites [45], however, in some cases, three magnetic sextets were used for the spectra fit, and two of them were associated with the B sites and denoted B1 and B2 (Table S1, SM) [46–48]. These sextets can be attributed to, e.g., low and high concentrations of nearest-neighbor cation substitutes in the iron local microenvironments or different probabilities of the number of substituted cations in the iron local microenvironment. Such substitutions also lead to line broadening and decreasing of the hyperfine magnetic field. Ion substitutions are also possible for iron in the tetrahedral site but only cause changes in the $B_{hf}$ value for related sexted [45]. Additionally, the ratio of the relative areas of components, which corresponds to $^{57}Fe$ in the B sites and in the A sites in the stoichiometric magnetite, is equal to two [49]. In some investigated annealed siderite samples (Table S1), this ratio is different from that expected. The effect of substitution can change the stoichiometric ratio. However, the observed nonstoichiometry of magnetite could also be explained in terms of the model proposed by Gorski and Scherer [50]. In the case of the fine particles of magnetite, one can expect surface defects that are predominantly the vacancies in octahedral positions. A decreasing of this expected ratio means an increase in vacancy concentration in this phase.

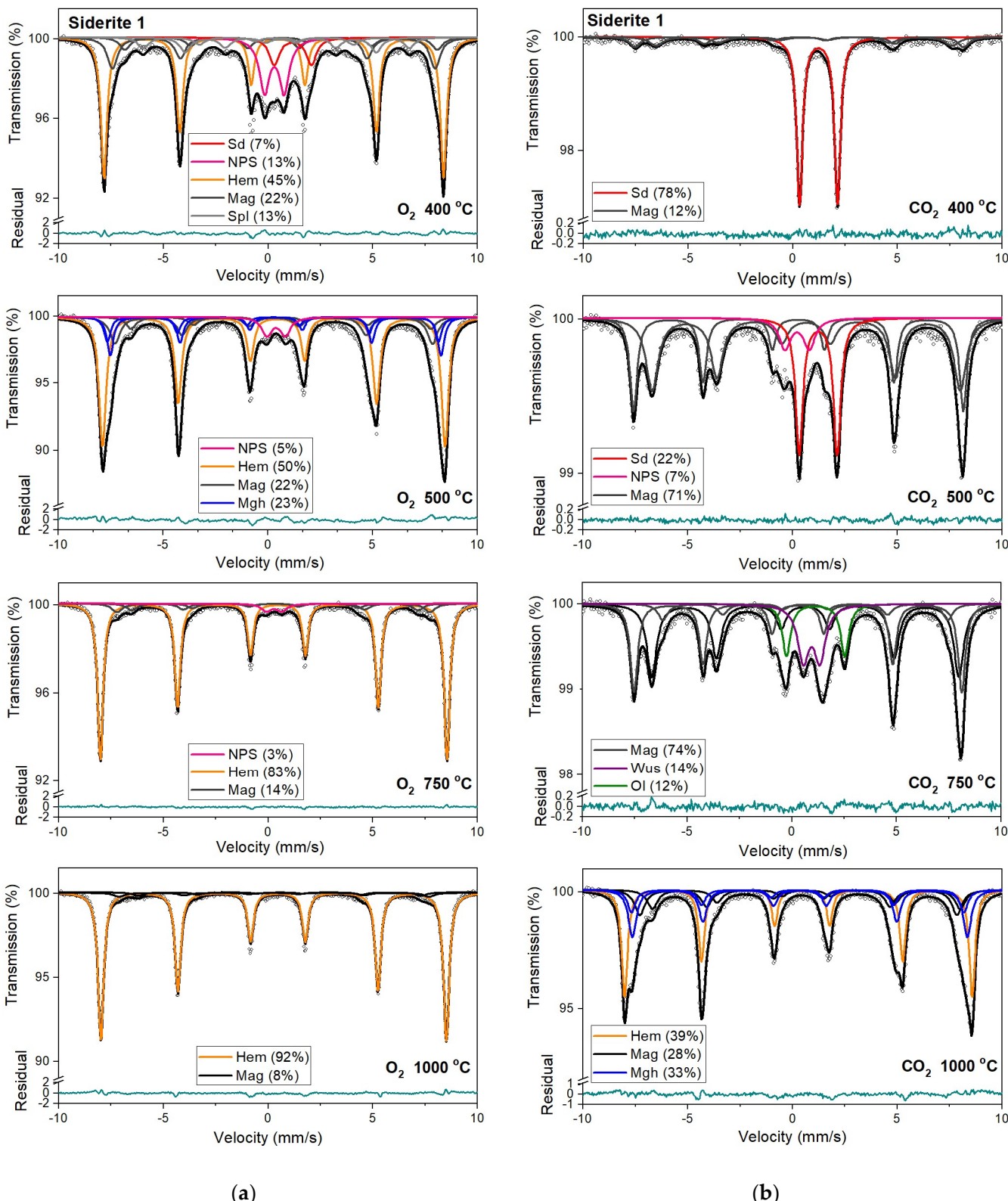

**Figure 6.** Mössbauer spectra of siderite sample S1 annealed at different temperatures in the $O_2$ (**a**) and $CO_2$ (**b**) atmosphere. Fitted subspectra (colored lines), their phase assignment, and contributions are shown on each spectrum. The residual is shown below each spectrum. Sd—siderite, NPS—iron oxides nanoparticles, Spl—spinel structure, Mag—magnetite, Mfr—magnesioferrite, Mgh—maghemite, Hem—hematite, Ol—olivine, Wus—wüstite.

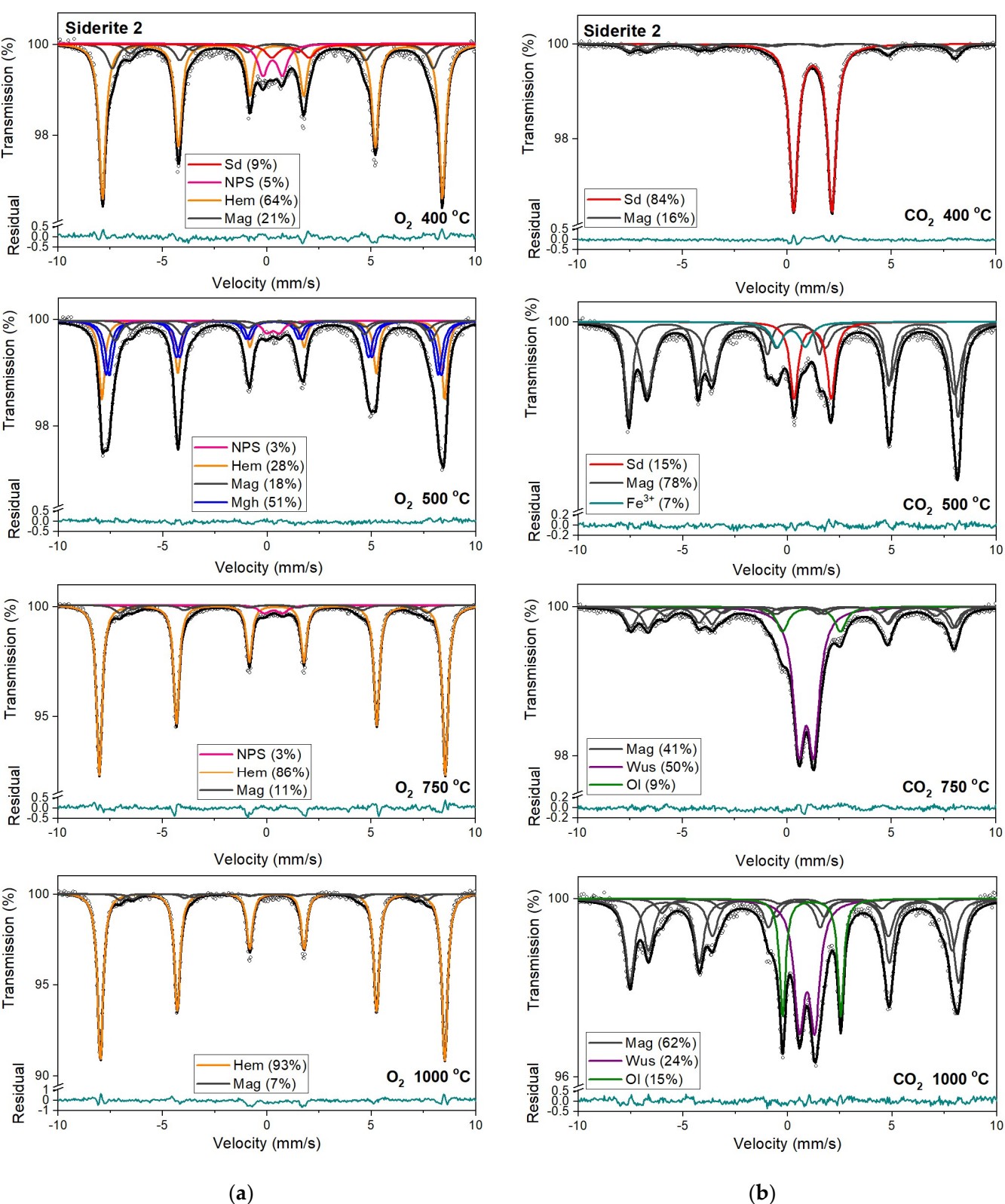

**Figure 7.** Mössbauer spectra of siderite sample S2 annealed at different temperatures in the $O_2$ (**a**) and $CO_2$ (**b**) atmosphere. Fitted subspectra (colored lines), their phase assignment, and contributions are shown on each spectrum. The residual is shown below each spectrum. Minerals abbreviations are the same as in Figure 6.

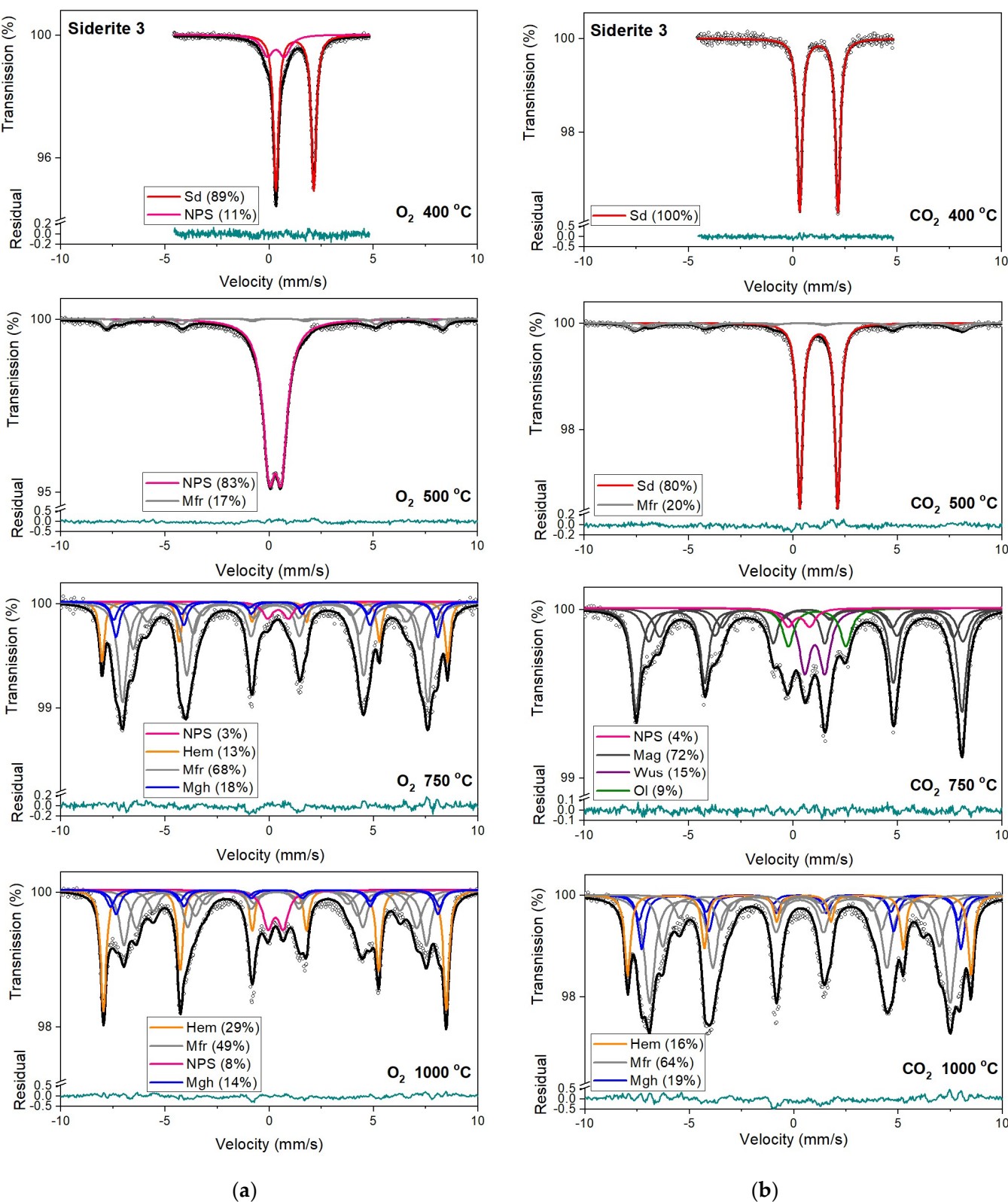

**Figure 8.** Mössbauer spectra of siderite sample S3 annealed at different temperatures in the $O_2$ (**a**) and $CO_2$ (**b**) atmosphere. Fitted subspectra (colored lines), their phase assignment, and contributions are shown on each spectrum. The residual is shown below each spectrum. Minerals abbreviations are the same as in Figure 6.

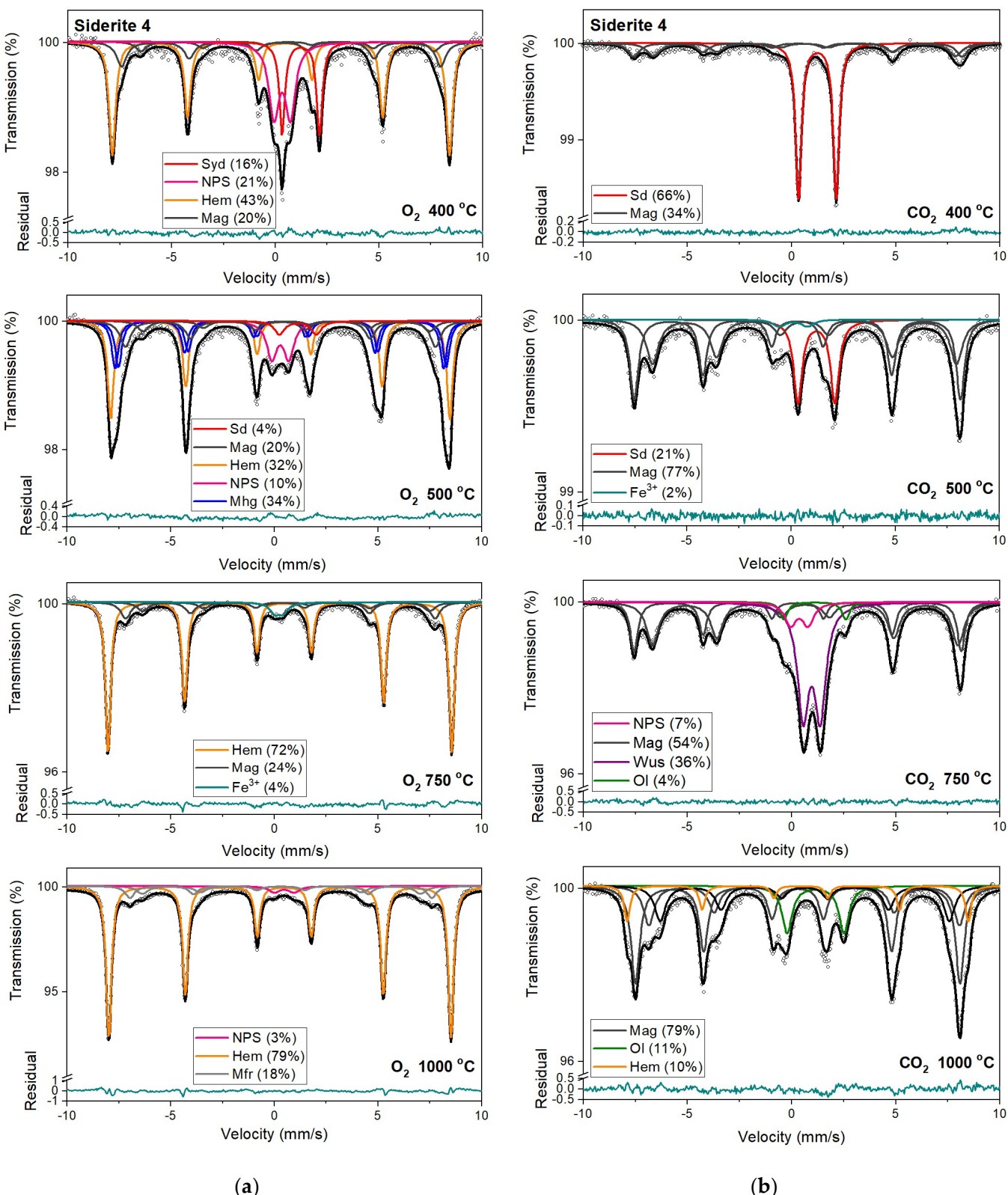

**Figure 9.** Mössbauer spectra of siderite sample S4 annealed at different temperatures in the $O_2$ (**a**) and $CO_2$ (**b**) atmosphere. Fitted subspectra (colored lines), their phase assignment, and contributions are shown on each spectrum. The residual is shown below each spectrum. Minerals abbreviations are the same as in Figure 6.

**Table 3.** Hyperfine parameters of sample S1 annealed at different temperatures (T) in the $O_2$ and $CO_2$ atmospheres.

| T (°C) | δ (mm/s) | Δ/2ε (mm/s) | $B_{hf}$ (T) | Γ (mm/s) | A (%) | Component |
|---|---|---|---|---|---|---|
| initial | $1.23 \pm 0.01$ | $1.80 \pm 0.01$ | - | $0.31 \pm 0.01$ | 100 | $FeCO_3$ |
| **Annealing in $O_2$** | | | | | | |
| | $0.89 \pm 0.01$ | $0.88 \pm 0.02$ | - | $0.60 \pm 0.04$ | 5 | NPS |
| | $0.37 \pm 0.01$ | $-0.17 \pm 0.01$ | $50.7 \pm 0.1$ | $0.40 \pm 0.01$ | 50 | $\alpha$-$Fe_2O_3$ |
| **500** | $0.25 \pm 0.02$ | $-0.01 \pm 0.01$ | $49.2 \pm 0.1$ | $0.38 \pm 0.02$ | 10 | (A) $\gamma$-$Fe_2O_3$ |
| | $0.38 \pm 0.01$ | $-0.01 \pm 0.01$ | $49.1 \pm 0.1$ | $0.38 \pm 0.02$ | 13 | (B) $\gamma$-$Fe_2O_3$ |
| | $0.29 \pm 0.01$ | $-0.02 \pm 0.01$ | $47.2 \pm 0.1$ | $0.50 \pm 0.02$ | 15 | (A) $Fe_3O_4$ |
| | $0.63 \pm 0.02$ | $0.02 \pm 0.01$ | $44.5 \pm 0.1$ | $0.60 \pm 0.05$ | 7 | (B) $Fe_3O_4$ |
| | $0.32 \pm 0.02$ | $0.76 \pm 0.03$ | - | $0.55 \pm 0.06$ | 3 | NPS |
| **750** | $0.37 \pm 0.01$ | $-0.20 \pm 0.01$ | $51.1 \pm 0.1$ | $0.31 \pm 0.01$ | 83 | $\alpha$-$Fe_2O_3$ |
| | $0.29 \pm 0.02$ | $0.00 \pm 0.01$ | $46.7 \pm 0.1$ | $0.55 \pm 0.04$ | 9 | (A) $Fe_3O_4$ |
| | $0.58 \pm 0.03$ | $0.00 \pm 0.01$ | $43.1 \pm 0.2$ | $0.60 \pm 0.05$ | 5 | (B) $Fe_3O_4$ |
| | $0.37 \pm 0.01$ | $-0.21 \pm 0.01$ | $51.3 \pm 0.1$ | $0.33 \pm 0.01$ | 92 | $\alpha$-$Fe_2O_3$ |
| **1000** | $0.28 \pm 0.03$ | $0.01 \pm 0.01$ | $46.3 \pm 0.2$ | $0.55 \pm 0.05$ | 4 | (A) $Fe_3O_4$ |
| | $0.56 \pm 0.03$ | $0.00 \pm 0.01$ | $42.1 \pm 0.2$ | $0.60 \pm 0.06$ | 4 | (B) $Fe_3O_4$ |
| **Annealing in $CO_2$** | | | | | | |
| | $1.23 \pm 0.02$ | $1.81 \pm 0.01$ | - | $0.40 \pm 0.01$ | 22 | $FeCO_3$ |
| **500** | $0.29 \pm 0.01$ | $0.88 \pm 0.06$ | - | $0.55 \pm 0.05$ | 7 | NPS |
| | $0.29 \pm 0.01$ | $-0.02 \pm 0.01$ | $49.0 \pm 0.1$ | $0.40 \pm 0.01$ | 33 | (A) $Fe_3O_4$ |
| | $0.65 \pm 0.01$ | $-0.01 \pm 0.01$ | $45.8 \pm 0.1$ | $0.60 \pm 0.01$ | 38 | (B) $Fe_3O_4$ |
| | $0.95 \pm 0.01$ | $0.72 \pm 0.01$ | - | $0.55 \pm 0.05$ | 14 | FeO |
| | $1.14 \pm 0.01$ | $2.78 \pm 0.01$ | - | $0.55 \pm 0.05$ | 12 | $(Mg,Fe)_2SiO_4$ |
| **750** | $0.29 \pm 0.01$ | $-0.02 \pm 0.01$ | $48.7 \pm 0.1$ | $0.42 \pm 0.01$ | 31 | (A) $Fe_3O_4$ |
| | $0.64 \pm 0.01$ | $-0.03 \pm 0.01$ | $46.1 \pm 0.1$ | $0.55 \pm 0.02$ | 30 | (B1) $Fe_3O_4$ |
| | $0.66 \pm 0.01$ | $-0.02 \pm 0.02$ | $43.4 \pm 0.2$ | $0.60 \pm 0.02$ | 13 | (B2) $Fe_3O_4$ |
| | $0.37 \pm 0.01$ | $-0.18 \pm 0.03$ | $51.6 \pm 0.1$ | $0.34 \pm 0.01$ | 39 | $\alpha$-$Fe_2O_3$ |
| | $0.25 \pm 0.03$ | $0.00 \pm 0.01$ | $49.3 \pm 0.1$ | $0.45 \pm 0.01$ | 11 | (A) $\gamma$-$Fe_2O_3$ |
| **1000** | $0.35 \pm 0.02$ | $0.00 \pm 0.02$ | $49.5 \pm 0.1$ | $0.45 \pm 0.01$ | 22 | (B) $\gamma$-$Fe_2O_3$ |
| | $0.28 \pm 0.01$ | $0.00 \pm 0.01$ | $47.1 \pm 0.1$ | $0.55 \pm 0.05$ | 16 | (A) $Fe_3O_4$ |
| | $0.66 \pm 0.02$ | $0.00 \pm 0.01$ | $45.7 \pm 0.1$ | $0.60 \pm 0.06$ | 12 | (B) $Fe_3O_4$ |

A quadrupole doublet related to siderite, with a relative area of ~4 %, is visible only for the S1 and S4 Mössbauer spectra after annealing the samples at a temperature of 500 °C (Table 3, Table S1). Magnetic sextets associated with hematite, magnetite, and maghemite ($\gamma$-$Fe_2O_3$) are present after annealing the samples at this temperature (except for S3). Maghemite is a cation-deficient spinel with $Fe^{3+}$ in both the A and B sites. The contributions from $Fe^{3+}$ in the A and B sites of magnetically ordered maghemite lead to the Mössbauer spectrum that consists of two sextets with similar $B_{hf}$ values of ~50 T (somewhat lower than that of hematite), essentially zero ε values [51] and different values of δ. Aluminum can be incorporated in the structure of maghemite [45,51] just as in that of hematite, leading to a systematic decrease in $B_{hf}$ values and line broadening.

Hematite with a more than 80% concentration is the main component of the Mössbauer spectra of samples S1, S2, and S3 annealed at 750 °C and 1000 °C (Table 3, Table S1). The

changing that occurs in sample S3 during its annealing looks similar to other samples. However, the temperatures where they take place are shifted to higher temperatures. What is more critical, magnesioferrite instead magnetite is visible in the Mössbauer spectra of this sample annealed at temperatures of 750 °C and 1000 °C. It is probably related to the high concentration of magnesium in the S3 sample. A similar effect of magnesioferrite formation in the fusion crust of ordinary chondrites was observed due to the presence of Mg from olivine and orthopyroxene [52].

Mössbauer spectra for samples annealed under the reducing atmosphere differ from those obtained in the oxidizing atmosphere. The thermal decomposition of siderite samples starts at a temperature of 400 °C, about one hundred degrees higher than in the oxidizing atmosphere. Just as in the case of annealing samples in an oxygen atmosphere, for sample S3, the process of siderite decomposition takes place at a higher temperature (500 °C). After annealing, the samples at a temperature of 400 °C, a quadrupole doublet related to siderite with a relative area higher than 60% is observed in the Mössbauer spectra. Also, two magnetic sextets related to $Fe^{3+}$ ions at the tetrahedral, and $Fe^{2.5+}$ ions at the octahedral, sites in magnetite are visible. One hundred degrees higher, the same components are seen in the Mössbauer spectra of all samples, but magnetite is the main iron oxide present in the samples. Magnetite, wüstite, and olivine appear as the main decomposition products of siderite after annealing the samples at 750 °C. Iron oxides like hematite, maghemite, magnetite, and magnesioferrite are the results of annealing the samples at a temperature of 1000 °C.

## 4. Discussion

X-ray diffraction, X-ray photoelectron spectroscopy, and Mössbauer studies show that all investigated carbonate rocks contain siderite as the only Fe-bearing mineral. Its content is more than 90% in samples S1 and S2 and about 85% in samples S3 and S4 (Table 1). Because siderite is the only iron-containing phase in all investigated samples, it is possible to follow the changes occurring in this mineral during annealing using Mössbauer spectroscopy. The iron content in all tested samples is about 37%, except in sample S3, where iron constitutes about 28% of the sample's volume (Figure 4) and contains much more Mg than other samples. The concentration of this element increases above 40% for all samples (for S3 is without changing) after annealing at a temperature of 500 °C and above 50% after annealing at 1000 °C. For S3, iron constitutes about 35% of the total volume.

All siderite samples were annealed at the temperature range from 200 °C to 1000 °C in the oxygen and under a $CO_2$ atmosphere. Both the temperature and the type of atmosphere affected the way of siderite decomposition. Thermogravimetric analyses and differential scanning calorimetry show that the initial temperature at which siderite decomposed in the oxygen was approximately 330 °C, while the final temperature was near 600 °C (except the S3 sample) with a maximum close temperature of 460 °C (Table 2, Figure 5). Under the $CO_2$ atmosphere, this process occurs at a temperature range of from 430 °C to 570 °C with a maximum of 500 °C. In oxidizing conditions, siderite decomposition begins at lower temperatures and occurs over a broader range of temperatures than under reductive conditions. For sample S3, the decomposition process of siderite is shifted to higher temperatures. The admixtures of other minerals like quartz or some clay minerals or substituting other ions instead of iron in the siderite structure can turn this process toward higher temperatures.

X-ray diffraction and Mössbauer measurements show that the decomposition of siderite in oxygen is a complex physicochemical process. Hematite is the final product of annealing the samples at higher temperatures ($\geq$750 °C), and the process of siderite decomposition at these temperatures can go off mainly according to the $4FeCO_3 + O_2 \rightarrow 2Fe_2O_3 + 4CO_2$ reaction [21]. The presence of hematite, magnetite, and maghemite as a result of annealing the samples at lower temperatures (<750 °C) suggests that the siderite decomposition can be described in a few steps at this temperature range. First, siderite decomposition produces a molecule of carbon dioxide and wüstite: $FeCO_3 \rightarrow FeO$

+ $CO_2$ [48,49]. Then, wüstite is oxidized according to the following equations: $4FeO + O_2 \rightarrow 2Fe_2O_3$ and $6FeO + O_2 \rightarrow 2Fe_3O_4$ [53]. The first step of siderite decomposition is so rapid that XRD and Mössbauer measurements detected only hematite and magnetite without wüstite presence. Additionally, the XRD and Mössbauer measurements clearly indicate the presence of maghemite in samples annealed at 500 °C. So, magnetite was transformed into maghemite and hematite according to equation $4Fe_3O_4 + O_2 \rightarrow 3\gamma\text{-}Fe_2O_3 + 3\alpha\text{-}Fe_2O_3$ [19]. As can be seen, the decomposition of siderite depends critically on heating temperature, time, and the availability of $O_2$. As was mentioned above, for the S3 sample, these processes are shifted to higher temperatures, and magnesioferrite is formed instead of magnetite due to the high concentration of magnesium in this sample.

The decomposition of siderite annealed under a $CO_2$ atmosphere can be described as a two-step function: $FeCO_3 \rightarrow FeO + CO_2 \rightarrow Fe_3O_4 + CO$ [53]. The first step of this reaction is very rapid, so only magnetite is visible in the XRD diffractograms and in the Mössbauer spectra of samples annealed up to a temperature of 500 °C. Annealing the samples at a temperature of 750 °C looks interesting. Due to this act, magnetite, wüstite, and olivine are present in the samples. What is important, wüstite is not a result of the reaction present above, but the $Fe^{3+}$ ions from magnetite are reduced to $Fe^{2+}$ to form wüstite. Then, wüstite reacts with quartz to create an Fe-rich silicate: $2FeO + SiO_2 \rightarrow Fe_2SiO_4$ [54]. The main products of annealing the samples S1 and S3 at 1000 °C are magnetite, maghemite, and hematite. The creation process of wüstite from magnetite and, consequently, olivine is still visible for sample S2. For sample S4, wüstite was not detected. It can suggest that in contact with quartz, even magnetite can form an Fe-rich silicate: $2Fe_3O_4 + 3SiO_2 \rightarrow 3Fe_2SiO_4$ [54].

## 5. Conclusions

The presented study shows the results of the thermal decomposition mechanism of siderite in an oxidizing $O_2$ and reducing $CO_2$ atmosphere. The phase analysis of iron-bearing minerals formed during these processes is shown. Most of the conclusions were drawn based on the results of Mössbauer spectroscopy. This technique represents a powerful tool to investigate the electronic structure, magnetic behavior, phase transitions, and bonding properties of numerous materials. As we know, in this method, we treat the [57]Fe nucleus as a probe of its local environment. It is a useful technique for giving quantitative information about hyperfine interactions of the [57]Fe nucleus. Our results show that both the temperature and the type of atmosphere affected the way of siderite decomposition, which is a complex process. Magnetite, hematite, and maghemite are products of siderite decomposition after annealing in the oxygen atmosphere at a temperature of 500 °C. In contrast, hematite is the main component of the samples detected after annealing at 1000 °C. Magnetite was present as the main compound in the samples annealed at 500 °C under a $CO_2$ atmosphere. However, magnetite, hematite, wüstite, and olivine were products of siderite decomposition at 1000 °C under this atmosphere. These processes lead to changes in iron concentration in samples. After annealing at a temperature of 500 °C, its concentration increases from about 37% to about 45%, and after annealing at 1000 °C, it is more than 50% of the total volume.

**Supplementary Materials:** The following supporting information can be downloaded at https://www.mdpi.com/article/10.3390/min13081066/s1, Table S1. Hyperfine parameters of siderite rock samples annealed at different temperatures (T) and in the $O_2$ and $CO_2$ atmospheres. Δ—isomer shift, Δ—quadrupole splitting, ε—quadrupole shift for magnetically split spectra, Bhf—magnetic hyperfine field, Γ—line width (full-width at half maximum), A—relative subspectrum area, (A)—tetrahedral and (B)—octahedral site. The maximal errors for the Mössbauer parameters were: ±0.03 mm/s for δ, ±0.06 mm/s for Δ/2ε, ±0.2 T for Bhf, ±0.06 mm/s for ε and ±1 % for A; Figure S1. Room temperature Mössbauer spectra of siderite sample S1 annealed at temperatures 400 °C in the oxygen atmosphere (a) and cooled to temperature of liquid nitrogen (b). Fitted subspectra (colored lines), their phase assignment, and contributions are shown on each spectrum. Sd—siderite, NPS—iron oxides nanoparticles, Mag—magnetite, Hem—hematite, Spl spinel structure. The residual is shown below each spectrum; Table S2. Hyperfine parameters of siderite sample S1 annealed at temperature (T) 400 °C in

the air and cooled to temperature of liquid nitrogen (LN). δ—isomer shift, Δ—quadrupole splitting, ε—quadrupole shift for magnetically split spectra, $B_{hf}$—magnetic hyperfine field, Γ—line width (full-width at half maximum), A—relative subspectrum area, (A)—tetrahedral and (B)—octahedral sites in magnetite. The maximal errors for the Mössbauer parameters were: ±0.03 mm/s for δ, ±0.06 mm/s for Δ/2ε, ±0.2 T for $B_{hf}$, ±0.06 mm/s for Γ and ±1 % for A.

**Author Contributions:** M.K.-G. did the conceptualization; designed the methodology; and performed the measurements/investigations, and validation. She also prepared the original draft and reviewed and edited the manuscript, secured the resources; methodology, M.K.-G. and J.N.; investigation, M.K.-G, J.N., M.S., J.K. and M.W.; formal analysis, M.K.-G., J.N. and M.S. All authors have read and agreed to the published version of the manuscript.

**Funding:** This research received no external funding.

**Data Availability Statement:** The data may be made available by the authors upon individual request.

**Acknowledgments:** Constructive comments of two anonymous reviewers and journal guest editor Michael Oshtrakh are gratefully acknowledged. This work was supported by the funds granted under the Research Excellence Initiative of the University of Silesia in Katowice. Special thanks to Tomas Kmjeć from Charles University in Prague (Czech Republic) for performing the Mössbauer measurement of sample S1 at the temperature of liquid nitrogen.

**Conflicts of Interest:** The authors declare no conflict of interest.

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
