# Peer review of "Thermal Decomposition of Siderite and Characterization of the Decomposition Products under O2 and CO2 Atmospheres"

_minerals, doi:10.3390/min13081066_

Round 1

Reviewer 1 Report

The work is relatively well presented. There is a problem with the fitting of the Mossbauer data of the magnetite phases as the relative areas of the subcomponents of the A and B sites are not what is expected for stoichiometric magnetite in many cases e.g. O2 anneal of S2/300, S4/400,  S1/500, S2/500, S4/500, and CO2 anneal of S1/400, S2/400, S4/400 etc. The B site area/intensity is correctly double the A site. This inconsistency must be explained/commented on.

The English is generally clear. In Figs 6-9 please do not obstruct the spectra with the legends. The phase legends need to be explained and corrected for uniformity. Please include y-axis units (%).

I list some minor corrections to aid clarity of some points.

59: "disproportionate into iron and magnetite" but there is no iron in the following equation (line 60)  ?

100: “raw”

106: “the phase content < 2%” - meaning ?

115: “used the”

175: “reflections”

222: “are in regard”

232-233 : “unchanging for S3”

236: “Elemental concentrations for samples annealed at 500…”

240: ” The TGA and DSC profiles..”

253: “to that (350C)..”

258: “by exotherms”

269: “the important fact for MS on siderite is…”

275: “in the MS of these..”

278: “samples may suggest the..”

324” “quadrupole”

387: “clearly”

388: “in samples”

397-401: rewrite. Meaning not clear.

408: “atmospheres”

Author Response

Comment 1

The work is relatively well presented. There is a problem with the fitting of the Mossbauer data of the magnetite phases as the relative areas of the subcomponents of the A and B sites are not what is expected for stoichiometric magnetite in many cases e.g. O2 anneal of S2/300, S4/400,  S1/500, S2/500, S4/500, and CO2 anneal of S1/400, S2/400, S4/400 etc. The B site area/intensity is correctly double the A site. This inconsistency must be explained/commented on.

Answer

Yes, I agree that this ratio of intensities of components, which corresponds to the contributions of Fe2.5+ ions in octahedral sites and iron Fe3+ ions in tetrahedral sites in the stoichiometric magnetite, is different from the expected. The observed nonstoichiometry of magnetite could be explained in terms of the model proposed by Gorski and Scherer. In the case of the fine particles of magnetite, one can expect surface defects – predominantly the vacancies in octahedral positions. Decrees of this expected ratio mean an increase in vacancy concentration in this phase. I added this explanation in the text.

Comment 2

I list some minor corrections to aid clarity of some points.

Answer

All the correctios were added in the text.

Reviewer 2 Report

The manuscript presents the results of the study of siderite thermal decomposition in an air atmosphere and under CO2 atmosphere. Siderite is the mineral that is the most abundant in sedimentary iron formation on Earth. Investigations were carried out by use of different methods such as X-ray diffraction, X-ray photoelectron spectroscopy, X-ray fluorescence method, differential scanning calorimetry and thermogravimetric analysis and spectroscopy measurements. This work is relevant and has practical significance.

I have some remarks concerning this manuscript however.

It is necessary to show in the figures the differential spectrum (difference between experimental and calculated spectra), because it is difficult to verify the correctness of the spectra processing. Besides there is no estimation of errors in the Mössbauer spectrum parameters.

One of the quadrupole doublets was related by the authors to iron-oxide nanoparticles (NPS). However, this doublet may also refer to particles of ferrihydrite, which may be X-ray amorphous. The presence of 83% nanoparticles in siderite sample S3 annealed at 500 K in the air seems very strange. Using the spectra measured only at room temperature, in my opinion, it is impossible to make a statement about the presence of iron oxide nanoparticles.

Author Response

Comment 1

It is necessary to show in the figures the differential spectrum (difference between experimental and calculated spectra), because it is difficult to verify the correctness of the spectra processing. Besides there is no estimation of errors in the Mössbauer spectrum parameters.

Answer

The residual is shown below each spectrum. I also added the maximum error values of the hyperfine parameters in Table 1S. If that's not enough, I'll add the exact uncertainties of each parameter.

Comment 2

One of the quadrupole doublets was related by the authors to iron-oxide nanoparticles (NPS). However, this doublet may also refer to particles of ferrihydrite, which may be X-ray amorphous. The presence of 83% nanoparticles in siderite sample S3 annealed at 500 K in the air seems very strange. Using the spectra measured only at room temperature, in my opinion, it is impossible to make a statement about the presence of iron oxide nanoparticles.

Answer

I agree that it is difficult to distinguish NPS from ferrihydrite or goethite based on room temperature hyperfine parameters of the quadrupole doublet. However, the last two can be excluded due to the specificity of the samples and measurements. Unfortunately, I do not have a spectrometer enabling measurements at low temperatures. However, one of the samples was measured at the liquid nitrogen temperature. The concentration of the component associated with the NPS decreased significantly at the expense of an increase in the concentration of hematite. I hope this evidence is enough. I added spectrum and parameters in supplementary materials.

Round 2

Reviewer 2 Report

Please add the residual  spectra on the Figure 1S.

Author Response

Dear Professor,

Thank you for your comments on my manuscript. I added the residual to the Figure 1S.

Sincerely

Mariola Kądziołka-Gaweł